# Rapalink-1 reveals TOR-dependent genes and an agmatinergic axis-based metabolic feedback regulating TOR activity and lifespan in fission yeast
Juhi Kumar, Kristal Ng & Charalampos Rallis ✉

The Target of Rapamycin, TOR, is a conserved signalling pathway with characterised chemical inhibitors such as rapamycin and torin1. Bi-steric third-generation inhibitors, such as rapalink-1 have been developed, however, their effects on organismal gene expression and lifespan have not been characterised. Here, we demonstrate that rapalink-1 affects fission yeast spatial and temporal growth and prolongs chronological lifespan with a distinct TORC1 selectivity profile. Endosome and vesicle-mediated transport and homeostasis processes related to autophagy render cells resistant to rapalink-1. Our study reveals TOR-regulated genes with unknown roles in ageing, including all fission yeast agmatinases, the enzymes that convert agmatine to putrescine and urea. Through genome-wide screens, we identify sensitive and resistant mutants to agmatine and putrescine. Genetic interactome assays for the agmatinase *agm1* and further cell and molecular analyses demonstrate that impairing the agmatinergic branch of arginine catabolism results in TOR activity levels that are beneficial for growth but detrimental for chronological ageing. Our study reveals the anti-ageing action of agmatinases within a metabolic circuit that regulates TOR activity, protein translation levels and lifespan.

Pharmacological inhibition of the evolutionarily conserved, nutrient-responsive and pro-ageing Target of Rapamycin signalling pathway presents great interest in disease and biogerontology[1,2]. The centrepieces of the pathway are the TOR PI3 kinase-related kinases that operate within two structurally and functionally distinct protein complexes, TORC1 and TORC2. The complexes are reported to have roles in both spatial and temporal aspects of cellular growth[3]. While TORC1 is responsible for promoting protein translation, lipid and central carbon metabolism and inhibiting autophagy, TORC2 regulates the cytoskeleton and is required for cell survival[3–5]. Rapamycin, a macrolide able to inhibit TORC1 in an allosteric manner through interaction with the FKBP12 protein (Fkh1 in fission yeast), is shown to extend lifespan in cellular and animal models[1,6–8]. Torin1, a dual ATP-competitive inhibitor of TORC1 and TORC2, has also been shown to be beneficial for lifespan in various models, including fission yeast[9]. Both drugs, together with several rapalogues, are used in numerous clinical trials ([www.clinicaltrials.org](www.clinicaltrials.org)).

Rapalink-1 is a third-generation bi-steric TOR inhibitor that combines rapamycin and sapanisertib (MLN0128)[10]. Its structure does not disrupt the binding of rapamycin to FKBP12 or the FRB domain of TOR. While

MLN0128 has been shown to have poor in vivo efficacy, its link to rapamycin has shown great promise in targeted TORC1 inhibition and cancer regression[10,11]. Rapalink-1 has been reported to prevent ethanol-induced senescence in endothelial cells[12]. Nevertheless, its precise effects on chronological lifespan (CLS) have not been addressed, and it is not known whether they will resemble rapamycin or an ATP-competitive inhibitor such as torin1.

Here, we have investigated the effects of rapalink-1 through comparisons with rapamycin using fission yeast, a relevant model in cell biology and ageing studies[13–15]. Both drugs advance mitosis in fission yeast, albeit with different profiles and kinetics, without halting the cell cycle like the pan-TOR inhibitor torin1[9] and can prolong CLS. The effects observed on gene expression following rapalink-1 treatment, are consistent with TOR[7–9,16] inhibition and new TOR-dependent genes are revealed. Using a combination of classical cell biology and genome-wide cell-based screens, we uncover new TOR-related genes and demonstrate that the enzymes converting agmatine to urea and the polyamine putrescine (agmatinases) are a pivotal pathway that positively impacts lifespan. Microbiota-derived agmatine has been previously reported to prolong host lifespan due to changes in lipid

Research Centre for Molecular Cell Biology, School of Biological and Behavioural Sciences, Queen Mary University of London, Mile End Road, E1 4NS London, UK.
✉e-mail: c.rallis@qmul.ac.uk

homeostasis. However, here, and for the first time, we reveal that the function of enzymes breaking down agmatine is linked to ageing. Analysis of genetic interactome of agmatinase *agm1*[17,18] together with cellular and molecular analyses, uncover a metabolic regulatory feedback loop that impacts on the regulation of TORC1 activity itself. Our data, reveal a more complex image around the agmatinergic function: while agmatine may affect metabolism and lifespan, agmatinases are pivotal in maintaining low TOR activity levels during TORC1-inhibition states, such as starvation or dietary restriction, with clear positive consequences on lifespan. Our data have possible implications to other organisms, including human cells and provide additional novel information on arginine metabolism and the role of polyamines[19,20] and polyamine-generating enzymes in ageing.

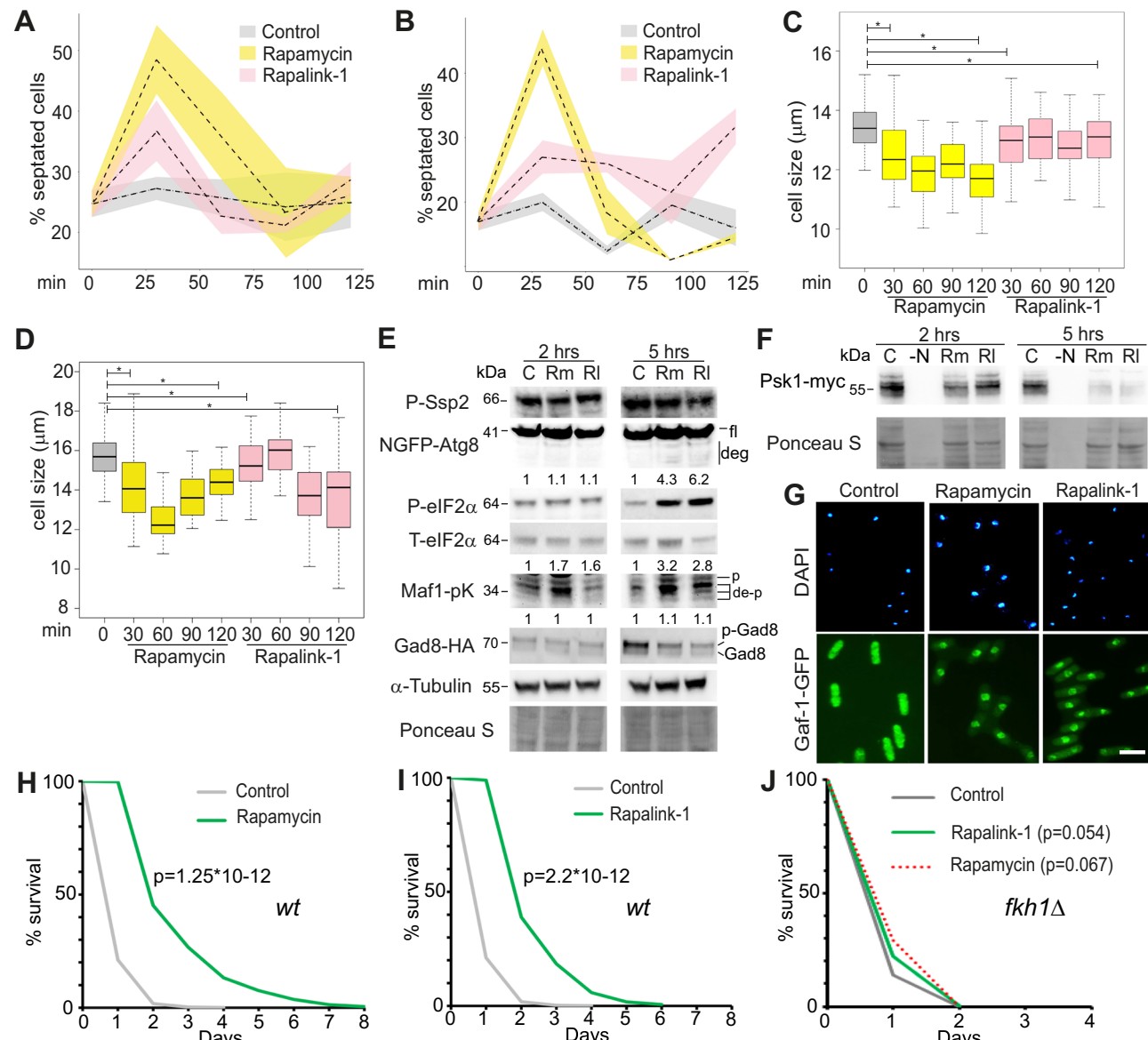

**Fig. 1 | Rapalink-1 affects TORC1-related processes in fission yeast and prolongs lifespan. A** Time course of septation index for untreated (control) and rapamycin and rapalink-1 treated cells (100 nM), as indicated, in YES media. Dotted lines represent average values while coloured ribbons standard deviation from three independent counts. **B** Same as in A in EMM2 minimal media. **C** Time course of cell size upon division measurements following rapamycin and rapalink-1 treatments in YES media. Asterisks indicate statistically significant differences between the compared groups ($p < 0.01$, wilcoxon testing). **D** Same as described in panel C with the assay conducted in EMM2 media. Asterisks indicate statistically significant differences between the compared groups ($p < 0.01$, wilcoxon testing). **E** Western blots for markers dependent on TORC1 activity and representative loading controls. Assays have been conducted following 2 and 5 h of treatments with 100 nM rapamycin (Rm) and rapalink-1 (Rl) together with untreated control cells (C). NGFP-Atg8 blot, fl: full length, deg: degraded forms; Maf1-pK blot, p: phosphorylated form, de-p: dephosphorylated forms; Gad8-HA blot, p-Gad8: phosphorylated form, Gad8: dephosphorylated form. Numbers above P-eIF2α blot correspond to ratios of phosphorylated versus total eIF2a normalised to the corresponding control (control will, therefore, always be 1). Numbers above Maf1-pK blot correspond to ratios of top (p) band to the second band and normalised with the corresponding control. Numbers above Gad8-HA blot correspond to ratios of the top (p-Gad8) to the bottom band (Gad8) and normalised with the corresponding control. **F** Western blots for Psk1-myc in control, C; nitrogen starvation, -N; rapamycin, Rm and rapalink-1, Rl treatments for 2 and 5 h. Assays have been conducted with the same drug concentrations as in (**E**). **G** DAPI staining and Gaf1-GFP localisation panels in control untreated, rapamycin and rapalink-1 treated cells as indicated. Bar is 10 μm. **H** CLS assays for untreated (control) and rapamycin treated wild-type fission yeast cells (log-rank $p = 1.25*10^{-12}$, see materials and methods). **I** CLS assays for untreated (control) and rapalink-1 treated wild-type fission yeast cells (log-rank $p = 2.2*10^{-12}$, see materials and methods). **J** CLS assays for untreated (control), rapalink-1 and rapamycin-treated *fkh1Δ* mutant cells (log-rank $p = 0.054$ for rapalink-1 and $p = 0.067$ for rapamycin, see materials and methods).

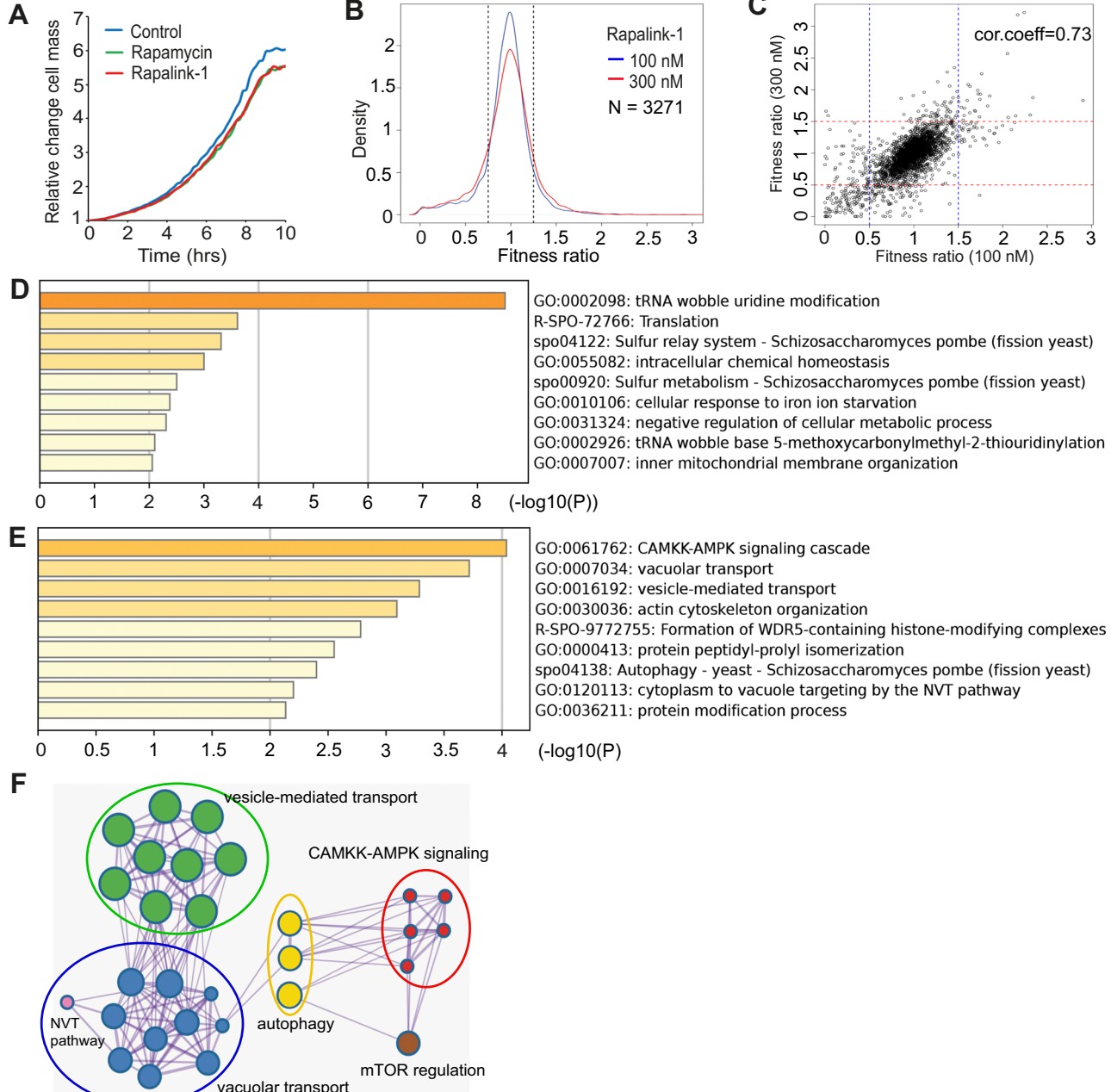

**Fig. 2 | A genome-wide screen for rapalink-1 points to the endosome-vacuole-lysosome network as important in TOR-dependent lifespan regulation.**
**A** Growth curves for control untreated, rapamycin and rapalink-1 treated wild type fission yeast cells, as indicated. The assay records light scattering dependent on culture cell mass (see materials and methods). **B** Fitness ratio overviews of genome-wide screens (data acquired for 3271 deletion mutants) for rapalink-1 at the indicated concentrations. Vertical dotted lines indicate cutoffs for sensitive and resistant mutants (Supplementary Data 2). **C** Fitness ratio correlation between the two conducted screens. Each point represents the average fitness obtained for a single deletion mutant of the library. **D** Gene ontology enrichment for mutants sensitive to rapalink-1 treatment. **E** Gene ontology enrichment for mutants resistant to rapalink-1 treatment. **F** Network of enriched terms coloured by cluster ID, where nodes that share the same cluster ID are typically close to each other.

## Results
### Rapalink-1 affects cell cycle progression, cell size and TORC1-dependent processes in fission yeast

Rapalink-1 is reported to preferentially inhibit TORC1[10,11] in human cells. We, therefore, wished to investigate its effects by comparing it with rapamycin, a well-characterised and widely used TORC1 inhibitor and without the complications of dual (TORC1 and TORC2) inhibitors such as torin1[21],[22]. Rapamycin-induced inhibition of TOR advances mitotic onset, mimicking the reduction in cell size at division observed following shifts to poor nitrogen sources[16]. We, therefore, analysed related effects when cells

are treated with 100 nM rapamycin and rapalink-1 as this concentration of rapamycin has been shown previously to inhibit TORC1-related processes in fission yeast[7,8]. In rich YES (Yeast Extract with Supplements) media, treatment with rapamycin and rapalink-1 leads to increased mitotic index (49 and 37%, respectively, compared to 27% in control cells), peaking at 30 min post treatments (Fig. 1A). Similarly, when cells are treated with rapamycin in EMM2 (Edinburgh Minimal Media 2) the mitotic index peaks at 30 minutes post-treatment (44% compared to 20% in control cells) returning to levels comparable with the untreated cells following this (Fig. 1B). Treatment with rapalink-1 leads to a milder but sustained increase

in the mitotic index (27, 26 and 32% at 30', 60' and 120' respectively). Along with mitotic onset, treatments with both rapamycin and rapalink-1 result in decreased cell size upon division in rich YES media (see 30'–120' time course in Fig. 1C). Rapamycin has a more severe effect on cell size compared to rapalink-1, at least in these media (Fig. 1C) with gradual decrease over time (from 30 to 120 min) while cells following rapalink-1 treatment decrease their size in 30 min and remain in the same levels for the whole examined period of 2 h (Fig. 1C). Likewise, in minimal EMM2 media, cell sizes upon division decrease for both rapamycin and rapalink-1 (Fig. 1D). The patterns of cell sizes differ between the two drugs: while in rapamycin cells reach their minimum size in 60 min (12.4 μm) and then start to gradually recover, in rapalink-1 they decrease in 90 min (13.5 μm) and remain at the same size range for the duration (120') of the assay (Fig. 1D).

Our septation and cell size results indicate phenotypes related to TORC1 inhibition as previously shown and described[7–9,16]. We, therefore, investigated whether drug treatments at these concentrations affect multiple aspects of TORC1 functions[4]. Negative cross-regulation between the AMPK and TORC1 pathways have been previously reported in fission yeast[23,24]. We, therefore, examined the phosphorylation status of the fission yeast serine/threonine protein kinase AMPK catalytic (alpha) subunit Ssp2. Following two and five hours of treatment with 100 nM of rapamycin or rapalink-1, no observable changes in Ssp2 phosphorylation are found (Fig. 1E). TORC1 negatively regulates autophagy[4]. Examination of GFP-Atg8 patterns[25] following drug treatment in the aforementioned time points, do not show a significant degree of processing[25] indicating that effects are not pronounced, at least in these timeframes (Fig. 1E). However, global protein translation might be suppressed five hours post-treatment with both drugs, as seen through increase in phosphorylation of eIF2α while its total levels remain unchanged at both of the 2 h and 5h time points, examined (Fig. 1E, numbers indicate the ratios of total to phospho-eIF2α). We wondered whether drug treatment could affect the activity of Maf1, a Pol III repressor[26], through its TORC1-dependent phosphorylation status[23]. Immunoblotting for Maf1-pK revealed that following treatment with both rapamycin and rapalink-1, Maf1-pK becomes dephosphorylated, more evidently in five hours of treatment (Fig. 1E, numbers indicate ratios of second band-a dephosphorylated form to top-phosphorylated form[23,24]). While the above results indicate effects on TORC1-dependent processes, we examined whether TORC2-dependent effects are evident following rapalink-1 treatments. Towards this, we have examined the phosphorylation status of Gad8, a kinase dependent on TORC2[8]. The ratios of phosphorylated to dephosphorylated Gad8 forms do not change in the treatments and timelines examined (Fig. 1E, numbers indicate the ratio of phosphorylated-top and dephosphorylated-bottom forms). To confirm effects of rapalink-1 on protein translation (Fig. 1E) we have analysed the levels of Psk1-myc (Psk1 is an S6 kinase[27]) following nitrogen starvation as well rapamycin and rapalink-1 treatments for two and five hours. In nitrogen starvation global translation and Psk1-myc drop drastically (Fig. 1F). Rapamycin (Rm) and rapalink-1 (Rl) treatments result in reduced amounts of Psk1-myc with the total protein content remaining in comparable levels with those of the control (Fig. 1F). This result strengthens the data, supporting that global protein translation is affected in rapalink-1 treatments as previously seen with rapamycin in fission yeast[8].

We have previously shown that the GATA transcription factor Gaf1 mediates Pol II- and Pol III-mediated transcription downstream of TORC1[9]. Gaf1 is localised mainly in the cytoplasm and upon TORC1 inhibition translocates to the nucleus and binds to target genes[9]. We examined localisation of Gaf1-GFP (controlled by the endogenous Gaf1 promoter as described in Rodríguez-López et al., 2020). Gaf1-GFP signal is found throughout the cell in untreated cells (Fig. 1G). Treatment with rapamycin and rapalink-1 results in nuclear localisation of Gaf1-GFP within 5 min (Fig. 1G) indicating that both drugs can inhibit TORC1 at sufficient levels to allow Gaf1 dephosphorylation and translocation. One of the major actions of Gaf1 is to repress all the Pol III transcribed tRNAs in fission yeast upon TORC1 inhibition. The result of Gaf1-GFP localisation agrees with Maf1 activation seen through Maf1-pK dephosphorylation (Fig. 1E).

Overall, our results show that rapalink-1 primarily targets fission yeast TORC1 and, like rapamycin, can inhibit several but not all aspects of TORC1-dependent functions in the examined concentrations. We cannot exclude the possibility that in higher concentrations or prolonged exposures to the drug, TORC2 will not be affected (as previously has been shown in the case of rapamycin[28]).

We then assessed the CLS of fission yeast cells when treated with rapamycin and rapalink-1 in YES media and at $OD_{600}$ 0.5 at 100 nM concentration. As expected, rapamycin induces CLS extension (Fig. 1H, logrank $p = 1.25*10^{-12}$). Similarly, rapalink-1 prolongs lifespan in similar fashion to rapamycin (Fig. 1I, $p = 2.2*10^{-12}$) demonstrating the potential of the drug and its action in ageing through TORC1 inhibition. Rapalink-1 lifespan extension depends on $fkh1$ (Fig. 1J, $p = 0.054$) as has previously been observed in the case of rapamycin[8] and observed here too (Fig. 1J, $p = 0.067$) demonstrating that the effects of rapalink-1 in these conditions are TORC1-dependent.

## A genome-wide screen for rapalink-1 resistance highlights roles of endosome-related processes in growth and lifespan

Previous data in fission yeast have shown that TOR inhibition results in lifespan extension with (in the case of torin1) or without (in the case of rapamycin) cell-cycle progress inhibition. Fast-growing cultures of wild-type fission yeast cells at $OD_{600}$ 0.2 were treated either with DMSO (control) or rapamycin and rapalink-1 at a 100 nM concentration. Cultures were placed at a Biolector setup as previously described[8] and growth was monitored continuously through light scattering measurements[7,8,29]. While no lag phase following drug treatment is observed (Fig. 2A), a mild growth effect is seen in this assay, consistent with results previously seen with rapamycin[8]. We then performed a genome-wide fitness screen for rapalink-1 using the Bioneer deletion library[30] at two rapalink-1 concentrations (100 and 300 nM, Fig. 2B) and at conditions where each mutant is assessed and measured in quadruplicate[7,9]. Results from the screens correlate well (Pearson cor.coeff = 0.73, Fig. 2C, Supplementary Data 2). Sensitive strains are related to protein translation, sulphur metabolism and mitochondrial membrane organisation among others (Fig. 2D) representing proteins and groups related to anabolic functions and cellular growth[31]. Resistant strains are related to CAMKK-AMPK signalling, vacuole and vesicle-mediated transport and autophagy (Fig. 2E). The fitness signatures acquired are consistent with TORC1 inhibition, with GO groups functionally linked to transport of macromolecules, processing of proteins, membranes and autophagic functions (Fig. 2F) representing fundamental metabolic functions in the crossroads of catabolism and growth[32]. Most of these mutants are required for normal lifespan[33], and they have been identified in previous TOR inhibition screens from our group[9] and others[34].

## Rapalink-1-dependent gene expression points to TORC1-related genes with both known and unknown roles in ageing

Beyond proteostasis, autophagy and global translation levels, TORC1 activity controls gene expression at the level of transcription through multiple transcription factors[5,35,36]. We therefore performed RNA-seq analysis on fission yeast cells treated with 100 nM rapamycin and rapalink-1 for 5 h. This timepoint was chosen based on the observed changes in the relevant markers described at Fig. 1E, F.

PCA analysis shows that rapalink-1 induces a stronger effect on gene expression compared to rapamycin with the PCA1 component accounting for 71.36% difference between control, rapamycin and rapalink-1 datasets separating well rapalink-1-treated transcriptomes from untreated and rapamycin-treated states (Fig. 3A). In these conditions, 17 genes are 2-fold, 70 genes 1.5-fold and 277 genes 1.2-fold upregulated in rapamycin-treated cells compared to untreated cells while 2, 4 and 87 genes are downregulated at the 2-fold, 1.5-fold and 1.2-fold levels respectively (Fig. S1A, Supplementary Data 3 and 4). In the case of rapalink-1-treated cells, 190 genes are 2-fold, 355 genes 1.5-fold and 701 genes 1.2-fold upregulated while 4, 28 and 516 are downregulated (Fig. S1B) compared to untreated cells (see

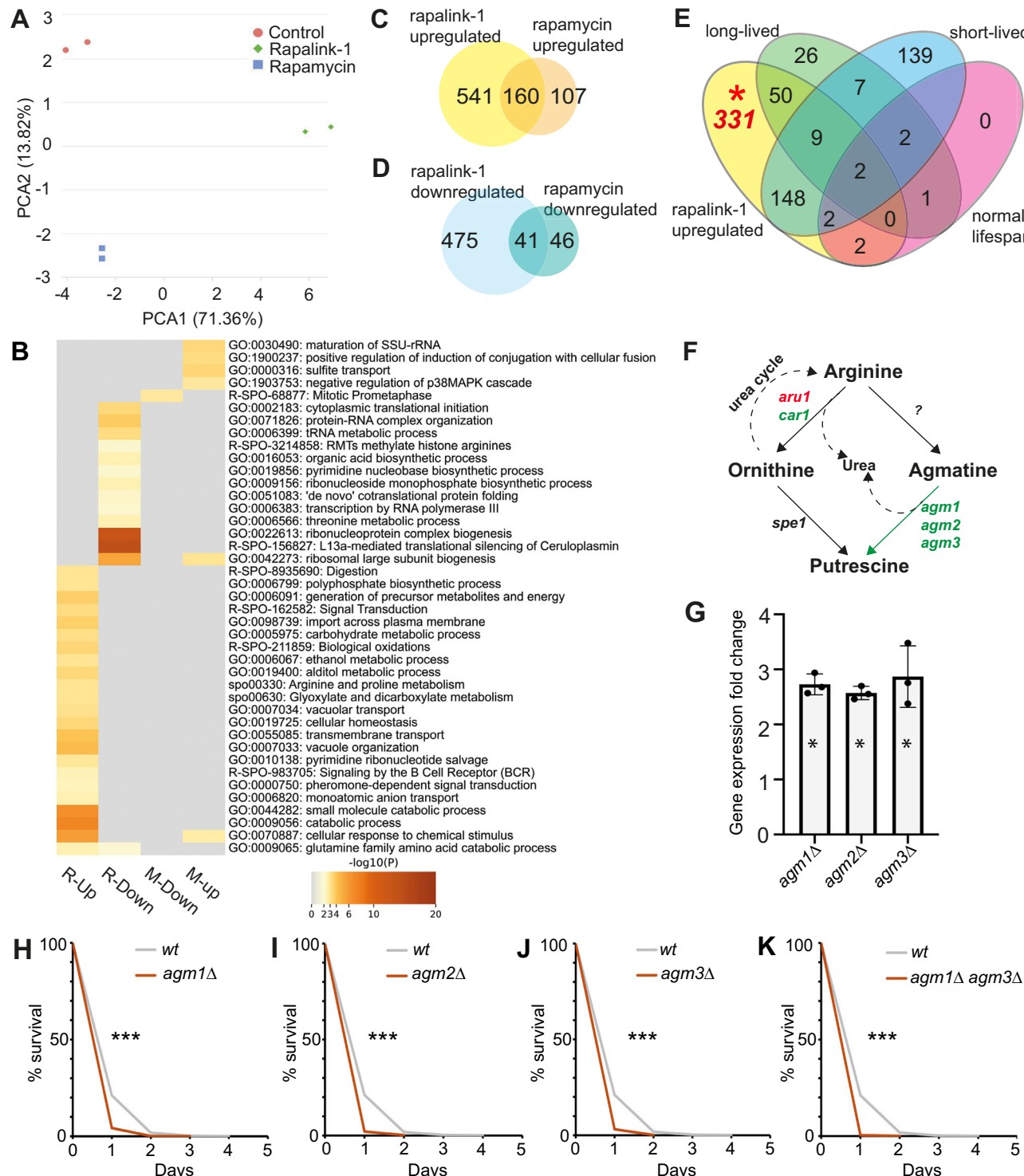

**Fig. 3 | Gene expression analysis following rapamycin and rapalink-1 treatment reveals novel genes involved in chronological ageing. A** PCA analysis of control untreated, rapamycin- and rapalink-1-treated cells as indicated. **B** Heatmap of representative GO enrichments for gene lists of upregulated and down-regulated genes in the drug treatments used (R-Up: rapalink-1 upregulated; R-Down: rapalink-1downregulated; M-up: rapamycin upregulated; M-down: rapamycin downregulated). **C** Venn diagram for upregulated genes following rapamycin and rapalink-1 treatments. **D** Venn diagram for downregulated genes following rapamycin and rapalink-1 treatments. **E** Venn diagram showing overlaps of rapalink-1 upregulated genes (the portion that corresponding mutants are viable) with genes that their deletions lead to long-lived, short-lived or normal-lived

mutants. **F** Schematic of arginine metabolism to ornithine, agmatine and putrescine with genes coding for responsible enzymes. In green: genes that are upregulated, red: downregulated, black: not affected, following rapalink-1 treatment. **G** qPCR validations of RNAseq data related to agmatinases. The bar graph shows fold-change of expression of the agmatinase genes *agm1*, *agm2* and *agm3* in fission yeast following five hours of rapalink-1 treatment compared to untreated controls. Single asterisks (*) within the bar graph indicates student test *p* value < 0.01 compared to controls. qPCR reactions were performed in triplicates.(**H-K**) CLS for normal (*wt*) and mutant *agm1Δ* (G, log rank *p* = 9.7 × 10⁻⁵), *agm2Δ* (H, log rank *p* = 1.5 × 10⁻⁵), *agm3Δ* (I, log rank, *p* = 3.9 × 10⁻⁵), and *agm1Δ agm3Δ* (J, log rank *p* = 3.3 × 10⁻⁶) cells as indicated. The asterists (***) in lifespans (H-K) indicate *p* values < 0.001.

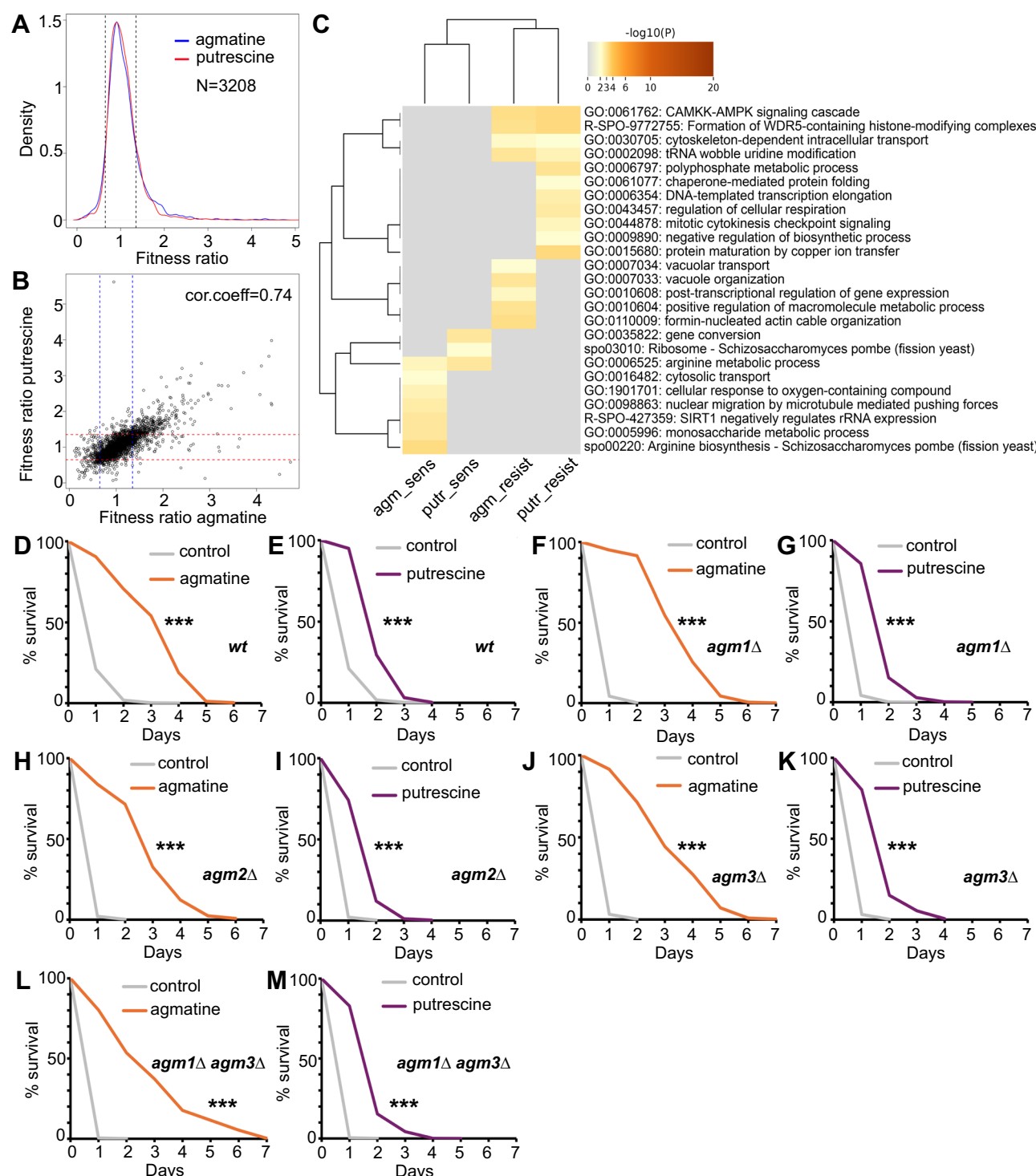

**Fig. 4 | Genome-wide fitness screens of agmatine and putrescine and effects on chronological lifespan of fission yeast. A** Density profiles of fitness ratios for 3208 *S. pombe* deletion mutants following agmatine and putrescine treatments. **B** Correlation of fitness ratios of deletion mutants in agmatine and putrescine screens. Each point represents the average fitness value for each mutant. **C** Heatmap of representative Gene Ontology enrichments for mutant strains resistant and sensitive to drug treatments (agm_sens: sensitive to agmatine; putr_sens: sensitive to putrescine; agm_resist: resistant to agmatine; putr_resist: resistant to putrescine). **D–M** CLS for normal (*wt*) and agmatinase mutant cells following agmatine and putrescine supplementation during growth phase (see material and methods) as indicated. The three asterisks (***) in lifespans indicate *p* values < 0.001, see log rank *p* values in the main text.

Supplementary Data 5 and 6). Interestingly, our analyses reveal 277 genes upregulated and 87 downregulated genes in rapalink-1 compared to rapamycin (Fig. S1C). Following rapamycin treatment, the downregulated gene list is enriched among others in cytoplasmic translation, tRNA metabolic process and transcription by polymerase III (Fig. 3B). Upregulated genes are

involved in cell cycle and specifically in mitotic prometaphase. On the other hand, the gene list of rapalink-1-upregulated genes is enriched in vacuolar transport and organisation. This has been previously observed in the case of torin1-treated fission yeast cells[9]. The subcellular localisation of TORC1 in fission yeast[37] is predominantly on the vacuole. It is interesting that factors

related to biogenesis and dynamics of the organelle where TORC1 resides are transcriptionally dependent on TORC1 activity. In addition, small molecule, amino acid metabolic processes and transmembrane transport are also enriched categories, among others (Fig. 3B). While both rapamycin and rapalink-1 target TORC1 and modulate lifespan in a similar fashion in terms of lifespan extension patterns, it may be that they do so through the regulation of common as well as distinct genes, at least, in the given analysed conditions. We, therefore, examined the overlaps of upregulated and downregulated genes for the two drugs as well as genes that are apparently regulated differently between the two pharmacological treatments. 541 upregulated (Fig. 3C) and 475 downregulated (Fig. 3D) genes are unique to rapalink-1. Genes that are upregulated in rapalink-1 compared to rapamycin are related to catabolism, small molecule catabolic processes, arginine and proline metabolism, glycolysis, transmembrane transport and small molecule catabolic processes (just to name a few, see Fig. S1D) while downregulated genes are related to ribosome biogenesis, translation, amino acid and tRNA metabolic processes (Fig. S1E). As catabolic processes, including autophagy, are over-represented in rapalink-1 overexpressed gene lists, we have examined whether rapalink-1 lifespan extension depends on autophagy mutants. We have examined the lifespans of *atg2Δ*, *atg8Δ*, *atg12Δ* and *atg22Δ* (atg mutants present within the Bioneer v5.0 deletion mutant collection[30]) without and with rapalink-1. Our results show that rapalink-1 can extend the CLS in all cases examined (Fig. S2A–D).

High-throughput phenotyping approaches have majorly contributed to characterising the CLS of haploid fission yeast mutants[33,35]. TORC1 is a major pro-ageing pathway, and TORC1-dependent genes are likely to be related in lifespan regulation. We, therefore, examined whether differentially expressed genes from rapalink-1-treated cells are linked with long or short reported lifespans of the corresponding mutants. While many rapalink-1 upregulated genes have reported lifespans, a significant number (331 genes) have no lifespan annotations (Fig. 3E). GO enrichment of these genes demonstrates that they are related to amino acid transport, amide transport, galactose, tyrosine, arginine and proline metabolism (Fig. S3A). Likewise, 150 genes that are found to be downregulated in rapalink-1 treated cells have not been annotated for lifespan (Fig. S3B) and these are related to L13a-mediated translational silencing, ribonucleoside biosynthesis, response to stimulus as well as to alanine, aspartate and glutamate metabolism (Fig. S3C). Examination of upregulated genes not linked yet to lifespan, revealed that arginine catabolism and specifically all the agmatinase enzymes involved in processing agmatine to putrescine are boosted following TORC1 inhibition (Fig. 3F). To validate the results from RNA-seq, we performed qPCRs for all three agmatinases from fast-growing cells ($OD_{600} = 0.5$) treated with rapalink-1 for 5 h compared to untreated cells. The results from qPCRs showed a 2.5- to 3-fold increase in all agmatinases, confirming the RNAseq data (Fig. 3G). We therefore wondered whether agmatinases are required for normal CLS in fission yeast. CLS assays reveal that *agm1Δ* (log rank $p = 9.7 \times 10^{-5}$, Fig. 3H), *agm2Δ* (log rank $p = 1.5 \times 10^{-5}$, Fig. 3I) and *agm3Δ* (log rank, $p = 3.9 \times 10^{-5}$, Fig. 3J) mutant cells are short-lived compared to *wt* cells. In addition, cells mutant for both *agm1* and *agm3* are also short-lived (log rank $p = 3.3 \times 10^{-6}$, Fig. 3K). Rapalink-1 prolongs CLS in *wt* cells (Fig. 1I). But does it have effects on the agmatinase mutants? We therefore performed CLS on single *agm* mutants with and without rapalink-1 treatments. In all cases rapalink-1 can extend lifespan (Fig. S4A–C), probably due to compensation between the agmatinase enzymes. The lifespan extension for *agm2Δ* is less pronounced compared to the ones observed for *agm1Δ* and *agm3Δ*. Nevertheless, the change is significantly different in all cases (log rank *p* value < 0.01). We are not able to say whether the differences in rapalink-1-dependent lifespan extension signifie specialisation in agmatinase requirements, but it is plausible.

## Agmatine and putrescine affect growth, metabolism and lifespan

The requirement of agmatinases in normal CLS led us to wonder about the effects of supplementing agmatine and putrescine on the fitness and lifespan of fission yeast cells. Towards this, we have performed genome-wide screens for both agmatine and putrescine(using 15 and 25 mM concentrations,

respectively). Each mutant has been assessed in quadruplicate and fitness ratios have been obtained through normalisations with untreated copies of the deletion library[30,38]. Good quality data have been obtained for 3208 mutants (Fig. 4A). Fitness ratios of the mutant strains between the two screens are well correlated (Pearson cor.coeff = 0.74, Fig. 4B) showcasing that they have similar metabolic and growth outcomes. Gene ontology enrichments for sensitive and resistant strains (cutoffs set at >1.35 and <0.65 fitness ratios for resistant and sensitive, respectively) show overlapping and unique enrichments (Fig. 4C, Supplementary Data 7 for agmatine and putrescine fitness ratios of all mutants). Both compounds render cells with gene deletions related to arginine metabolism, as sensitive. However, while putrescine affects more strongly ribosome-related mutants, agmatine affects mutants related to cytosolic transport or response to oxygen-containing compound (Fig. 4C for a complete list of GOs). On the other hand, mutants related to the CAMKK-AMPK signalling cascade (as in the case of rapalink-1) and intracellular transport and tRNA wobble uridine modification are resistant to both agmatine and putrescine. The list of resistant strains to putrescine is enriched in categories related to polyphosphate metabolic process, cellular respiration and chaperone-mediated folding, among others (Fig. 4C). Interestingly, GO enrichments for strains resistant to agmatine include vacuolar transport and organisation and are reminiscent of lists of mutant strains resistant to rapalink-1 (genes coding for endolysosomal-related proteins). This result together with the requirement of agmatinase enzymes in normal fission yeast CLS led us to wonder whether agmatine and putrescine supplementation can be beneficial to *S. pombe* CLS. Indeed, addition of both compounds to fast growing wild-type *S. pombe* cultures at the same concentrations used in our genome-wide screens lead to lifespan extension with agmatine having more profound effects (Fig. 4D, log rank $p = 5.3 \times 10^{-11}$; Fig. 4E $p = 9.4 \times 10^{-10}$). Our results are consistent with previous observations on the positive effects of agmatine in lifespan and ageing[39]. We examined whether agmatinases are required for the lifespan extension and repeated the experiments in the genetic background of *agm1Δ* (Fig. 4F, log rank $p = 4.4 \times 10^{-13}$; Fig. 4G, $p = 4.5 \times 10^{-10}$), *agm2Δ* (Fig. 4H, log rank $p = 3.5 \times 10^{-11}$; Fig. 4I, $p = 3.5 \times 10^{-8}$) *and agm3Δ* (Fig. 4J log rank $p = 1.1 \times 10^{-13}$; Fig. 4K, $p = 2 \times 10^{-9}$), gene deletions. Lifespan extensions comparable with the *wt* cells are observed in all single mutant backgrounds. The same beneficial effects are also observed in double *agm1Δ agm3Δ* mutant cells (Fig. 4L log rank $p = 4.1 \times 10^{-10}$; Fig. 4M, $p = 3 \times 10^{-10}$). This might be due to either functional compensation or beneficial effects related to other pathways such as lipid metabolism[39] or through oxidative stress protection[40].

## *Agm1* interactome reveals metabolism-mediated TOR activity regulation with effects on growth and lifespan

To understand the mechanisms underlying the requirements of agmatinases' functions in ageing we have defined the genetic interactome of *agm1* by performing a synthetic genetic array (SGA). In this assay, we systematically construct double mutants: the mutant of interest with each of the mutants contained in a whole deletion library. Colony sizes for single and double mutants are used as proxies for their fitness. A negative genetic interaction is scored when the fitness of the double mutant is worse than expected based on the individual mutants' fitness. Examples are synthetic lethality and synthetic sickness. Negative genetic interactions can be observed between genes of parallel pathways that contribute to the same cellular function. A positive interaction is scored when the fitness of the double mutant is unexpectedly high (again, based on the fitness of the corresponding single mutants). Positive interactions can reveal suppression (one mutation compensates for the defect caused by the other) or epistasis (one mutation masks the effect of another in a pathway) relationships. We have used *ade6::natMX6* as a control SGA query. As the deletion library strains carry *ade6* auxotrophy alleles[30,38], the control query/SGA does not affect the fitness profiles of the deletion collection as previously reported in published methods[7]. Following normalisations for plate positional effects, low fitness mutants, and linkage we have isolated data for 3,108 double mutants with *agm1* (Supplementary Data 8). Physical mapping of the isolated genetic interactions is reassuring of the successful double mutants'

generation as it demonstrates the characteristic accumulation of negative values around the *agm1* locus and the positive values around the control *ade6* locus (Fig. 5A, blue and red lines indicate the physical location of query gene *agm1* and the control query *ade6* in the *S. pombe* genome). These values have been filtered out during our quality control steps. With interaction value cutoffs of -15 and +15 (see materials and methods) we find 179 mutants showing negative interactions and 284 mutants exhibiting positive interactions with *agm1* (Fig. 5B). The negative interaction list is enriched for genes coding for post-transcriptional regulation, carbon metabolism, and reticulophagy among other GO terms (Fig. 5C, Supplementary Data 8). Positive interactions are enriched in amide metabolism, stress response and nitrogen metabolism (Fig. 5D, Supplementary Data 8).

Close inspection of the interactors' list reveals that genes coding for stress and amino acid sensing that inhibit TORC1 such as *gcn1*[41] and *fil1*[42] positively interact with *agm1* (Fig. 5E, green fonts). On the other hand, the TORC1 core component *tco89*[3], negatively interacts with *agm1* (Fig. 5E, red fonts). These results could mean that the absence of *agm1* (and possibly the rest of the agmatinases) promotes growth that is enhanced through the absence of stress response players. The cellular growth enhancement is TORC1-dependent as absence of *tco89* results in reduced cellular fitness/growth. Therefore, in this scenario, agmatinases may provide a metabolic regulatory feedback for TORC1 activity in fission yeast: when TOR is inhibited, cells process vacuole-stored amino acids such as arginine[43], thus ensuring that TOR inhibition signals are maintained, relayed within the cell and preventing anabolism while preparing for catabolic processes and autophagy. In this case, *agm* mutants would showcase phenotypes related to increased TORC1 activity. Indeed, we have found that all agmatinase deletion mutants have short chronological lifespans compared to *wt* cells (Fig. 3H–K). *Agm1* mutants have already been reported to be fast-growing[44] and to have decreased mating efficiency[45], both cellular characteristics of increased TORC1 activity. We have then assessed cell size upon division for normal (*wt*) and agmatinase mutants. Our results demonstrate that *agm* mutant cells are larger (Fig. 5F, ***: $p < 0.001$, Wilcoxon testing) strengthening the case of increased TORC1 levels as the latter controls both temporal and spatial aspect of fission yeast growth. TORC1 activity growth through protein translation via phosphorylation of S6 kinases (such as Sck1, Sck2 and Psk1). Indeed, phospho-S6 activity is increased in *agm* mutants, almost 5-fold for *agm1Δ*, 3.4-fold for *agm2Δ* and 1.7-fold for *agm3Δ* (Fig. 5G). In addition, P-eIF2α, a negative marker of translation also used for assessing 'stress' levels of cells, is reduced in *agm* mutants (Fig. 5G, 0.5-fold in *agm1Δ*, 0.1-fold for *agm2Δ* and 0.3-fold for *agm3Δ*). Interestingly, while *tco89* mutant cells are long-lived[8], *agm1Δ tco89Δ* are short-lived as single *agm1* mutants (Fig. 5H). This result shows that in the case of the CLS phenotype, *agm1* is epistatic of *tco89*. Our data reveal that the agmatinergic branch of arginine catabolism is favoured during TOR inhibition and forms a regulatory loop required for maintaining a lower TOR level, according to what environmental pressures dictate (Fig. 5I).

## Discussion

In this study, we have assessed the cellular and molecular effects of the bi-steric TOR inhibitor rapalink-1 using fission yeast, a genetically tractable eukaryotic model. Our results show that rapalink-1 primarily targets TORC1, prolongs chronological lifespan and demonstrate that TOR activity negatively regulates agmatinase genes. In fast-growing conditions and nutrient availability, TOR is highly active. The cells retain TOR activity, thus enhancing anabolism and promoting growth and, therefore, repressing agmatinase genes and the catabolic processing of arginine pools. This metabolic circuit ensures that arginine, stored within vacuoles[43], continues to provide TOR activation input. In stress conditions (including stationary phase and during CLS) and when TOR activity is lower[46], cells catabolise arginine. Once again, this ensures that in such conditions, cells upregulate defensive mechanisms, scavenge and recycle nutrients and prevent anabolism. Fission yeast arginases Aru1 and Car1 process arginine to ornithine.

Interestingly, upon direct TORC1 inhibition through rapalink-1 the main arginase Aru1 is downregulated while Car1 is upregulated 1.4-fold. Data from the orfeome and global localisation study in fission yeast[47] indicate that Aru1 and Car1 are localised in the cytoplasm and the nucleus. Nevertheless, Agm1 and Agm3 (no relevant data on Agm2 are available) are found in the endoplasmic reticulum (note that reticulophagy is a Gene Ontology term for genes interacting with *agm1*) and the vacuole. Therefore, agmatinase enzymes are localised in the right compartment to control arginine pools crucial for TORC1 activity and fission yeast cells' survival during nutritional and other stresses as well as in the context of lifespan. Taken together, these data and circuits can explain the requirements of the agmatinases in chronological ageing.

Our gene expression analyses have revealed many TORC1-regulated genes with no annotations or characterisation related to ageing and lifespan. Recent studies have focused on broad profiling of fission yeast mutants[35] using non-competitive settings. In addition, we and others have performed genome-wide assays[13] towards characterising mutants related to lifespan regulation[33,44]. Despite these efforts and due to the competitive nature of pooled barcode-based assays, only a fraction of the available viable mutants have been characterised. In addition, many medium- or high-throughput lifespan approaches have already been described and conducted in fission yeast[29,33,48]. Nevertheless, this gap of knowledge remains. Uncovering novel TORC1-regulated genes will focus current and future efforts in discovering new biology within the bio-gerontology field and beyond.

Parallel to gene expression analyses, we have performed cell-based genome-wide screens to reveal mutants that are sensitive and resistant to rapalink-1, as we have previously reported for torin1[9]. Our data showcase that the beneficial effects of TOR inhibition are related to the endolysosomal pathway and involve ESCRT, HOPS, CORVET, autophagy and the PIK3 complex. Taken together, these results may indicate that these complexes may be required for TOR-dependent lifespan control through inhibitors including rapalink-1, rapamycin and torin1. Recent work has linked the inhibition of S6K with lifespan increase through the endolysosomal system[49] showcasing the universal roles of these functional connections in the ageing process.

Interest in agmatine as a nutraceutical is significant, and it is reported as a promising therapeutic agent for treating a broad spectrum of central nervous system-associated diseases[50]. A large portion of agmatine in mammals is supplemented from diets and gut microbiota, and long-term (5 years) supplementation studies[51] have demonstrated its safety in the examined doses. Beneficial effects on the host's lifespan through microbiome-derived agmatine induced by metformin treatment have been previously reported[39,52]. Indeed, agmatine can augment metformin's therapeutic effects. However, it does not always promote beneficial effects as it can contribute to pathology[53]. Here we show that a consequence of the direct TORC1 inhibition is the transcriptional activation of all the enzymes that catabolise agmatine to putrescine and urea, with this pathway being vital for viability in non-dividing states. We have performed genome-wide phenomic screens revealing mutant backgrounds that benefit (or not) from agmatine and putrescine treatments. Our results show that supplementations of agmatine and putrescine are beneficial for cellular fitness only when arginine metabolism pathways are intact. This may explain diverse outcomes of agmatine supplementation and possible contributions to pathological conditions that are observed in other studies. Interestingly, both agmatine and putrescine prolong lifespan in *wt* and agmatinase mutants. Nevertheless, the effects of agmatine are more pronounced compared to putrescine, probably because agmatine can contribute to cellular health in multiple ways, as previously reported, including protecting from mitochondrial dysfunction and promoting the balance between mitochondrial fission and fusion[54].

There is a growing interest in polyamines and how they regulate longevity. The triamine spermidine, a natural polyamine, can be generated from both the ornithine and the agmatine axes. In mammalian cells its biosynthesis starts from arginine with the enzyme arginase (ARG1)

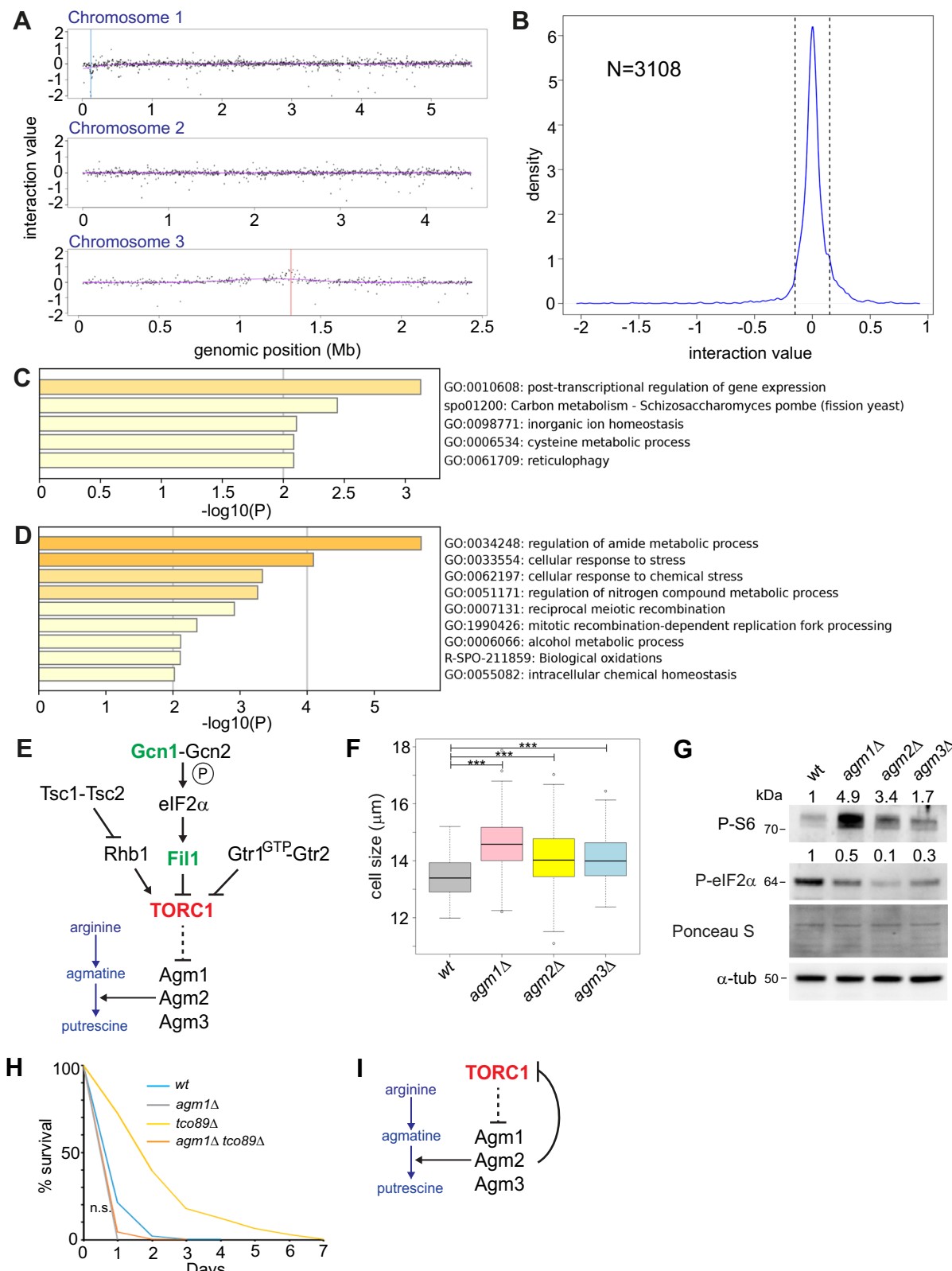

catalysing the reaction from arginine to ornithine[55]. Spermidine is well-known to stimulate cytoprotective mechanisms and autophagy[56]. Spermidine has been implicated with lifespan in yeast and we have previously uncovered longevity alleles and revealed connections of Rim15/Ppk31 with spermidine metabolism[19]. Recently, spermidine is shown to be pivotal in mediating fasting-induced autophagy in yeast, nematodes and human

cells[20]. Nevertheless, in our present study, we show that agmatinases do not function simply to contribute to putrescine and ultimately spermidine production. While spermidine induces autophagy and mediates hypusination of the translation regulator eIF5A[20], agmatinases' induction due to lowering of TORC1 activity levels contribute towards maintaining the low TORC1 activity. Our analyses reveal a metabolic feedback loop mediated

**Fig. 5 | Genetic interactomes of *agm1* reveal a metabolic feedback mechanism that tunes TORC1 activity. A** Physical mapping of *agm1* interaction values with 3108 mutants. Blue and red lines indicate the physical location of the query gene *agm1* and the control query gene *ade6* in the *S. pombe* genome. **B** Density plot of the genome-wide *agm1* genetic interactions. Vertical dotted lines show the cutoffs set for the interactome analysis (see materials and methods). **C** Bar graph representation of GO enrichments for negative *agm1* interactions. **D** Bar graph representation of GO enrichments for positive *agm1* interactions. **E** Schematic showing a previously reported tripartite TORC1 regulation in fission yeast. Green fonts signify positive while red fonts negative interactions. **F** Cell size upon division measurements in normal (wt) and *agm* mutant cells (*agm1Δ, agm2Δ, agm3Δ*) in YES media. Asterisks indicate statistically significant differences between the compared groups (p < 0.01, wilcoxon testing). **G** Western blot for P-S6 and P-eIF2α in normal (wt) and *agm* mutant cells (*agm1Δ, agm2Δ, agm3Δ*) with accompanied α-tubulin western blot and Ponceau S stains as loading controls. Numbers on top of P-S6 and P-eIF2α are ratios of signals to the corresponding tubulin signal and normalised to the *wt* ratio (thus ratio of *wt* is always 1; measurements were performed using ImageJ). **H** CLS patterns of *agm1Δ and agm1Δ tco89Δ* mutants is indicated. n.s.: not significant as log rank *p* > 0.05. **I** A schematic model for the revealed role of agmatinase enzymes in tuning TORC1 activity in fission yeast.

through agmatinase enzymes, ensuring that TORC1 activity levels are maintained as required for physiological cell needs and according to the signals received from the environment (nutritional or pharmacological). The revealed metabolic control is required for survival in non-dividing states and in CLS and can have implications and be of importance for human cells. Understanding how TORC1 activity is tuned may be beneficial in both normal ageing and also pathological states as well as in cancer where TOR plays important roles[2,57].

## Methods

### Strains and media

*972 h−* was used as the *wild-type*. The strain containing C-terminally GFP-tagged Gaf1 (*gaf1-GFP kanMX6*) has been generated according to[58] with *gaf1* being under the control of the endogenous promoter and as in refs. [7,9,59]. Strains Maf-1-pK and GFP-atg8 have been previously described[23,25]. We have also used *h+ agm1::kanMX6; h+ agm2::kanMX6; h+ agm3::kanMX6; h+ fkh1::kanMX6; agm1::natMX6 agm3::kanMX6; agm1::natMX6 tco89::kanMX6*[30]. Other strains have been obtained from the deletion collection and verified using PCR according to manufacturer's instructions. Microscopy was performed using an EVOS M5000 and Leica DMRA2 epi-fluorescent microscope fitted with a monochrome Orca-ER camera from Hamamatsu Photonics. YES (Yeast extract with supplements) and EMM2 media (Edinburgh Minimal Media 2) (Formedium) are used as indicated in the figures and main text. Drug treatments have been conducted in fast-growing cultures at $OD_{600}$ 0.5 (see main text for details). Liquid cultures were grown at 32 °C with shaking at 130 rpm.

### Western blotting

Cells were treated with 100 nM rapamycin or 100 nM rapalink-1 for 2 or 5 h as indicated in figures and main text. Cells were disrupted using RIPA buffer and glass beads in a FastPrep 5 G MP lysis system. Antibodies directed against phospho-eIF2a (#9721, 1:1000), eIF2a (#9722, 1:1000), phospho-Ssp2 (#50081, 1:1000), Phospho-(Ser/Thr) Akt Substrate Antibody (#9611, 1:2000) were purchased from Cell Signalling Technologies. The antibody directed against myc (ab32, 1:1000) was purchased from Abcam while antibodies directed against V5 (sc-81,594, 1:1000) and HA (sc-7392, 1:1000) were purchased from Santa Cruz Biotechnology. Secondary antibodies Goat Anti-Mouse IgG H&L (HRP) (ab205719, 1:5000 in all cases except when V5 and GFP was the primary antibody where it was used at 1:10,000) and Goat Anti-Rabbit IgG H&L (HRP) (ab205718, 1:5000) were purchased from Abcam while the ECL Western Blotting Detection system was from Pierce™. α-tubulin (T5168, 1:1000) and anti-GFP (11814460001, 1:1000) were purchased from Sigma-Aldrich. α-tubulin and Ponceau S staining have been used as loading controls for western blots as and where indicated. Western blotting quantifications have been performed using ImageJ/Fiji[29,60].

### Cell size and septation index determination

To measure septation index, cells were resuspended in calcofluor solution (#18909, Sigma-Aldrich) and incubated at room temperature for five minutes. Cells were visualised using an EVOS M5000 microscope. Percentages of septation from 200 cells for each time point have been recorded. For cell length at division, measurements of 50–100 septated cells were conducted using Fiji/ImageJ software[7,60].

### CLS assays

CLS was determined through counting colony-forming units every day and normalising numbers with day zero[8]. Fission yeast cells are grown to the stationary phase. However, in this study, cells are left for an additional 48 hours as seen in previous related studies[15] and then the assay starts with this time point considered as the 100% survival for the time course. Two or three independent biological repeats with each repeat having three technical replicates have been used. Log rank tests are performed for all CLS assays with CFUs and percentages used in the statistical testing. Statistical significance has been also validated with AUC measurements[60] as also seen in ref. 9. Lifespan lengths vary largely in yeasts depending on the time point used following stationary phase entry and the media used. Different batches of YES media can generate different lifespan curve patterns depending on the Yeast Extract used in the particular batch. We have used a unique YES batch (Formedium) throughout the study.

### RNA sequencing data

Untreated as well as rapamycin and rapalink-1 treated cells (treatment was at $OD_{600} = 0.5$) for 5 h were harvested and processed for RNA isolation[8] using acidic phenol-chloroform extraction method followed by RNAeasy (Qiagen) cleanup with on column DNase treatments before double elution. Following appropriate quality control steps, RNA library formed by polyA capture and Illumina 150 bp paired-end sequencing followed by standard R-based bioinformatic analysis was performed. Differential gene expression analysis of two conditions/groups (two biological replicates per condition) was performed using DESeq2R package (1.20.0). Genes with an adjusted *P* value <=0.05 found by DESeq2 and at various fold cutoffs (see main text) were assigned as differentially expressed. Sequences are deposited at GEO (accession number: GSE272269).

### Quantitative PCRs

As in the case for RNA sequencing untreated as well as rapalink-1 treated cells (at $OD_{600} = 0.5$, treatment for 5 h) were harvested and processed for RNA isolation[8] using acidic phenol-chloroform extraction method followed by RNAeasy (Qiagen) cleanup with on column DNase treatments before double elution. cDNA was prepared using LunaScript® (NEB) while qPCR reactions were conducted on a QIAquant96 2plex cycler (Qiagen) using Luna® (NEB) according to the manufacturer's instructions. Fold-changes of expression were calculated using the Ct method and normalising with alpha-tubulin 2 (*atb2*). Primers used are: agm1L1: 5'-TTTTGGAGGC GGCAAATCAA-3'; agm1R1: 5'-CCAACTCTGTCACGGATCCT-3'; agm2L1: 5'-TACTGTGCTTCCTCGAGTCC-3'; agm2R1: 5'-CCATCTGC TTCATCGCCATC-3'; agm3L1: 5'-TGATCAACAACGGCACATCC-3'; agm3R1: 5'-GCCAATCCAGGATCGACAAC-3'; atb2L1: 5'-TTCTGTG TATCCGGCTCCTC-3'; atb2R1: 5'-AGAGGCGGTGATGGAAGAAA-3.

### High-throughput phenomics screens

The deletion library[30] was arrayed on solid YES media at 1536-spot density, with each strain represented by four spots with various concentrations of drugs as indicated for each assay. Plates were incubated at 32 °C and high-resolution images of the plates were acquired. Colony size quantitations were then performed using the R package Gitter[61]. Median colony sizes were calculated for each plate and replicate. Strain colony size data per condition were normalised to the corresponding growth on YES.

## Synthetic genetic arrays

SGAs were performed using the Bioneer v5.0 haploid library[30] with each interaction examined in quadruplicate[7]. Query strains were the following: *h − agm1::natMX6*, and control query *h− ade6::natMX6*. Colony size was used as a proxy for double mutant fitness. Colony size measurements were obtained using the gitter package[61]. Library mutants that fell within 30 kb distance of query mutations were excluded from the respective datasets to avoid spurious interactions as well as mutants for which the *ade6* double mutant colony size was less than 50 pixels as they were deemed to be present in too little amount for accurate interaction values to be calculated. Colony sizes were normalised to the plate median accounting for the effect of the query mutation and plate-specific effects and normalised for column- or row-specific effects. Medians of colony sizes of the SGAs were normalised with respect to the *ade6* SGA (which represents fitness of library single mutants as this query does not affect the library) to calculate the value of genetic interactions between query and library mutants. Interactions with high within-replicate variability were excluded. Finally, the logarithms in base 10 of the interaction were used for interaction scores.

## Enrichment analyses of interactions

Enrichment analyses were performed using Metascape[62]. p-values were corrected for multiple tests according to FDR. Enrichment analysis was conducted by comparing lists of interacting genes to all genes in the dataset (Bioneer v5.0 collection strains[30]).

## Statistics and reproducibility

CLS assays have been performed in two or three biological repeats with 3 technical repeats for each biological repeat. Statistical differences were determined using log-rank tests. Cell sizes have been determined through measurements of 200 cells, while statistical differences were determined using Wilcoxon testing. qPCRs were performed in three biological replicates while statistical differences are determined using student's *t* test. SGA data have been collected through quadruplicates for each mutant. Processing and normalisations are described in the corresponding paragraph of the Materials and Methods. p-values for RNAseq and Gene ontology enrichments have been determined through standard methods and were corrected for multiple tests according to FDR.

## Reporting summary

Further information on research design is available in the Nature Portfolio Reporting Summary linked to this article.

## Data availability

RNA-seq data are deposited and are available from GEO (accession number: GSE272269). The Numerical data used for the generation of lifespans and qPCR graphs in the manuscript provided in Supplementary Data 1. Processed omics data (RNA-seq, genome-wide drug screens and SGA screens are all provided in Supplementary Data 2–8. All yeast strains are available from the corresponding author (or other sources/laboratories that have been generated in) on reasonable request. Uncropped western blot images are provided into the Supplementary Information (Supplemental Fig. 5).

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

## Acknowledgements

We thank members of the Rallis Lab Rowshan Islan and John-Patrick Alao for critical reading and valuable comments on the manuscript. We are grateful to the Bahler lab for the use of the Biolector micro-fermentation system used in Fig. 2A. We are grateful to Peter Thorpe for the use of the Singer RoToR HDA system. This work was supported by funding to C.R. from the Biotechnology and Biological Sciences Research Council [Research grant numbers: BB/V006916/1, BB/V006916/2]. C.R. also acknowledges support and funding of the group from the Medical Research Council [Grant number: MR/W001462/1]).

## Author contributions

Conceptualisation: C.R. Methodology: C.R. Investigation: J.K., K.N., and C.R. Formal Analysis: C.R. and J.K. Writing: C.R. Funding acquisition: C.R. Supervision: C.R.

## Competing interests

The authors declare no competing interests.
