## [Transparent Peer Review file · Communications Biology]

Rapalink-1 reveals novel mTOR-dependent genes and an agmatinergetic axis-based metabolic feedback regulating mTOR activity and lifespan

Corresponding Author: Dr Charalampos Rallis

Version 0:

Reviewer comments:

Reviewer #1

(Remarks to the Author)

The authors investigate the effect of RapaLink-1 in fission yeast. RapaLink-1 is a bivalent mTOR inhibitor consisting of a rapamycin-FRB binding element linked to a TOR kinase inhibitor. The authors claim that phenotypes of RapaLink-1, such as chronological lifespan extension, are consequences of TORC1 inhibition. The authors identify mutants that are either RapaLink-1-sensitive or resistant and characterize the transcriptomic changes induced by RapaLink-1 or by rapamycin. The authors notice that the expression of the 3 genes involved in processing agmatine to putrescine (agmatinases) increases upon RapaLink-1 treatment. Single mutant strains lacking agmatinases display a short lifespan compared to WT cells. Finally, the authors propose a model in which agmatine degradation is favored upon TORC1 inhibition to eventually negatively regulate TORC1. Specific comments are as follows.

1. The authors refer to TOR from any species (including fission yeast) as “mechanistic TOR”, abbreviated as “mTOR”. mTOR is also commonly used to refer to mammalian TOR only. Thus, referring to fission yeast TOR as mTOR is confusing since mTOR is also known as mammalian TOR. To avoid needless confusion, it would be very helpful if the authors referred to fission yeast TOR as “TOR” and not “mTOR”. In fact, the authors name the fission yeast TOR complexes as TORC1 and TORC2 (without “m”).
2. RapaLink-1 was originally shown to inhibit both mammalian TORC1 and TORC2 in MCF-7 cells (PMID: 27279227). The authors focus on the effect of RapaLink-1 in fission yeast on TORC1 but seem to give little relevance to the potential effect on TORC2. Does RapaLink-1 inhibit TORC2 in fission yeast?
3. Figure 1E. Data do not fully support the authors’ conclusions, i.e., changes in eIF2alpha and Maf1 phosphorylation upon treatment with RapaLink-1. The authors should replace these blots by other blots of better quality (lower exposure may help) or, at least, include semiquantitative measurements of band intensities.
4. Figure 1I. In this manuscript, the authors extensively focus on comparing the effects of rapamycin and RapaLink-1. They should ideally assess the effect of both drugs in cells lacking FKH1 on the same experiment. Now, the authors perform the experiment with RapaLink-1 and rely on their own previous result (2013) for rapamycin.
5. To assess TORC1 activity in cells lacking agmatinases, the authors use a phospho-AKT antibody which, as they mention, is a readout for several AGC kinases. One of them is Gad8, which is a substrate of TORC2 (PMID: 24928510). Thus, the readout that the authors use to claim that lack of agmatinases activates TORC1 (Figure 1J) is not specific for TORC1. The authors should assess TORC1 in cells lacking agmatinases using more specific TORC1 readouts as described in Figures 1E and F.
6. Figure 5I. The authors should ideally compare the effect of single *agm1*, *tco89* and double *agm1 tco89* mutations on the same experiment (similar concern as in comment 4).
7. Line 33. mTOR is a PI3 kinase-related kinase (also known as PIKK) and not a PI3-related kinase.

8. Related to Figure 3, lines 180 and 185. Figure 2B should be 3B. Figures 2C and 2D should be 3C and 3D.

Reviewer #2

(Remarks to the Author)

The manuscript by Kumar and Rallis is interesting and technically well performed.

I have a few points and suggestion for experiments:

-Avoid specific jargon, since this is a journal for non-specialized readers: Define rich YES media, introduce pik3, etc.

-The authors show convincingly that Rapalink-1 lifespan extension depends on fkh1
But does it depend on autophagy? The experiment should be repeated using ATG knock outs. I acknowledge that the authors did not observe enhances autophagy upon Rapalink treatment but subtle changes in autophagic flux may have been overlooked by the assay used.

Is spermidine enhanced upon Rapalink treatment? See Hofer et al, 2024, AUTOPHAGY? What about GABA concentrations?

Also the authors exclude this in the discussion: Does spermidine supplementation in Agmatine knock outs has the same effect as Putrescin supplementation on lifespan extension?

Reviewer #3

(Remarks to the Author)

Rallis review by Charles Hoffman

This study investigates the effect of the third-generation mTOR inhibitor rapalink-1 on the fission yeast *Schizosaccharomyces pombe* that has been used to study Tor signaling and the impact of rapamycin on *S. pombe* lifespan. This study demonstrates that rapalink-1 acts through TORC1 and goes on to identify biological processes are affected by rapalink-1 by examining changes in gene expression and by identifying strains from viable haploid deletion collection with increased or reduced sensitivity to rapalink-1. This work further uncovers a role in lifespan for agmatinases, a class of enzymes for which there had been relatively little research. Additional studies identified mutant strains that are sensitive or resistant to either agmatine or putrescine supplementation, and demonstrated that such supplements increase CLS in *S. pombe*. Finally, the authors examine more phenotypes associated with agm gene deletions and tie this all together in a model in which TORC1 regulates agmatinase activity, which also involves a feedback loop to regulate TORC1 activity. Taken together, this study provides novel and interesting insights into the effects of rapalink-1 expands our understanding of the roles of TORC1 in lifespan. Given the role of TORC1 in both growth and quiescence, this paper should be of broad interest.

I have some modest suggestions for improving the manuscript as well as some questions that should be addressed.

Supplementary Tables 1, 6 and 7 do not include common gene names, while Tables 2, 3, 4, and 5 do. Is there a reason for the difference? The addition of actual gene names would make these tables more accessible for the reader.

There are two sets of asterisks in Figure 1E that require explanation in the figure legend. I assume that for Maf1, these are presumed to represent phosphorylated and dephosphorylated forms, however the lower bands are so weak, I am not convinced that they are Maf1 versus a contaminating signal. What is the evidence that these bands represent dephosphorylated Maf1? Are these bands missing from a Western of a maf1 deletion strain? Does the protein shift to this lower band if the extract is treated with a phosphatase such as lambda protein phosphatase?

The increase in CLS comparison in Figure 2H is discussed in absolute values. I suggest discussing this with a statistician to see whether it is more appropriate to assess the difference in the fold-increase. While this is still greater in wild-type cells (3.4 fold) than in pik3D cells (3.025-fold), I do not know if this is statistically significant or if that difference is important to any of the conclusions.

While not necessary, it would be nice for readers who are not experienced in SGA analyses to include a sentence in the Results section explaining what a negative interaction versus a positive interaction looks like in the assay.

The authors need to clarify their use of asterisks with regard to statistical significance. Figure 1D has one, two and three asterisks without an explanation of their meaning. Elsewhere, they only use three asterisks, and do explain that. Figure 5H seems like something that would warrant an assessment of statistical significance.

For Figure 5H, are these technical or biological replicates?

Minor typos

Line 177- remove open parenthesis

Line 221- oxygen-containing compounds

Line 253- replace "to show" with "showing" to create parallel structure with "exhibiting"

Line 277- "seems is enhanced"- please clarify this sentence

Lines 280-281- "but does not reflect, in this case, to the ageing phenotype"- please edit

Line 294- ensures

Reviewer #4

(Remarks to the Author)

In this manuscript, Kumar and Rallis explore the effect of rapalink-1, a third generation bi-steric inhibitor of mTOR, on chronological lifespan of fission yeast. The authors' data suggest that rapalink-1 prolongs chronological lifespan of fission yeast cells with affecting expression of numerous genes. Among them, all of genes encoding agmatinases, which metabolize agmatine into putrescine, was upregulated in the presence of rapalink-1, and the deletion of those compromised lifespan. By performing genetic interactome analysis of agmatinase *amg1* together with cellular and molecular analyses, the authors found that the deletion of the agmatinases genes resulted in increased cell size and TORC1 activity. Based on these observations, the authors proposed a metabolic feedback circuit regulating TORC1 activity mediated by agmatinases. Although these findings provide new insights into the regulation of TORC1 signaling by agmatinases, some specific points need to be addressed prior to publication to strengthen the authors' conclusion.

Specific points

1) In Fig. 1E, TORC1 activity was monitored after 2 and 5 hours of rapamycin as well as rapalink-1 treatment. However, as the mitotic index is significantly increased even after 30 min of treatment with TORC1 inhibitors (Fig. 1A and B), the authors should assess the TORC1 activity at earlier time point (e. g. 30 min after treatment). Previous literatures have demonstrated that Maf1 is effectively dephosphorylated in the presence of rapamycin (Du et al., 2012, *Biol. Open*, 1, 884; Morozumi et al., 2021, *J. Cell Sci.* 134, jcs258865). On the other hand, only a tiny portion of Maf1 was dephosphorylated in authors' experiments, suggesting a possibility of mild suppression of the TORC1 activity. To carefully assess the TORC1 activity in these experimental conditions, another TORC1 substrate, such as Psk1 (much easier to detect its phosphorylation state than Maf1 (Nakashima et al., 2012, *J. Cell Sci.* 125, 5840; Morozumi et al., 2021, cited above)) should be examined.

2) The authors state that the up regulation of genes related to vacuolar transport and organization by rapalink-1 resembles the observation treated by torin1, an ATP-competitive inhibitor of both TORC1 and TORC2 (p8, lines175-177). Thus, it is likely that rapalink-1 is able to inhibits not only TORC1 but also TORC2 in fission yeast. Indeed, rapalink-1 can suppress both mTORC1 and mTORC2 activity in mammalian cells (e.g. Rodrik-Outmezguine et al., 2016, *Nature* 534, 272). Thus, authors should examine whether rapalink-1 inhibits TORC2 in fission yeast by monitoring the phosphorylation of TORC2 substrate such as Gad8.

3) The RNA-seq analysis shown in Fig. 3 revealed that the expression of agmatinase genes are increased upon rapalink-1 treatment. As the authors focused on these genes and performed further experiments, this observation should be confirmed by individual RT-qPCR.

4) In Fig. 5G, although the authors visualized the phosphorylation of TORC1 target kinases by using anti p-Akt targets antibody, it remains possible that those obtained bands are just non-specific signals. The authors should eliminate this possibility by including the control experiment under TORC1 inhibition, such as nitrogen starved or rapamycin-treated conditions.

5) By combining the results shown in Fig. 3G-H and Fig. 5, the authors proposed that shorter lifespan of the *amg* mutant cells compared to wild-type is due to the increased TORC1 activity. To further confirm this possibility, lifespan of the *amg* mutants in the presence of TORC1 inhibitor such as rapamycin and rapalink-1 needs to be tested.

Minor points

7) Fig. 1A, B: Dot lines are too thin, being difficult to see (particularly in rapamycin treated cells). Thicker lines should be used.

8) In Fig. 1E, the authors demonstrated that rapamycin and rapalink-1 treatment did not induce autophagy by monitoring the cleavage of GFP from Atg8. This result is consistent with previous observations (Takahara et al., 2012 *Genes Cells*, 17 698; Morozumi et al, 2021 cited above). However, the authors state autophagy may be up regulated (line 98-). This does not reflect the observation in Fig. 1E, and thus, should be eliminated.

8) Fig. 1E: Information about what asterisks indicate should be included.

9) Line 121: The P value of the result in Fig. 1H is "0.047" in the main text and figure legend, while "0.054" in the figure.

Which is true?

10) Lines 174, 180, 185: Cited figures should be Fig. 3 but not Fig 2.

11) Line 177: Parenthesis should be eliminated.

12) Fig. 5A: Please mention what the blue and red lines indicate.

13) Among the kinases listed in line 277, Gad8 is a direct substrate of TORC2 but not TORC1, and thus, should be removed from the sentence.

Version 1:

Reviewer comments:

Reviewer #1

(Remarks to the Author)

The authors have made a significant effort and have successfully addressed all issues raised by the reviewer.

Reviewer #2

(Remarks to the Author)

good revision

Reviewer #3

(Remarks to the Author)

The authors have fully responded to my questions and suggestions regarding the narrative, figures and statistical analyses. I am satisfied with the revised manuscript.

Reviewer #4

(Remarks to the Author)

The authors fully answered my concerns raised during review of the initial submission, and therefore, I recommend the manuscript for publication.

We deeply appreciate the reviewers' constructive criticism and useful comments. We have performed additional experiments and have amended the manuscript as suggested.

Below is a point-by-point answer (in normal font) to reviewers' comments (in bold font):

REVIEWER 1

1. The authors refer to TOR from any species (including fission yeast) as “mechanistic TOR”, abbreviated as “mTOR”. mTOR is also commonly used to refer to mammalian TOR only. Thus, referring to fission yeast TOR as mTOR is confusing since mTOR is also known as mammalian TOR. To avoid needless confusion, it would be very helpful if the authors referred to fission yeast TOR as “TOR” and not “mTOR”. In fact, the authors name the fission yeast TOR complexes as TORC1 and TORC2 (without “m”).

We have removed the term ‘mTOR’ and replaced it with ‘TOR’ following the suggestion and to avoid possible confusions.

2. RapaLink-1 was originally shown to inhibit both mammalian TORC1 and TORC2 in MCF-7 cells (PMID: 27279227). The authors focus on the effect of RapaLink-1 in fission yeast on TORC1 but seem to give little relevance to the potential effect on TORC2. Does RapaLink-1 inhibit TORC2 in fission yeast?

This has also been raised by other reviewers. Towards this:

We have tested the phosphorylation status of Gad8, a direct target of TORC2. We have used the previously published Gad8-HA strain PMID: 23551936. The results show that in the conditions used (2 and 5 hrs timeframes and rapalink-1 concentration-100 nM), rapalink-1 does not influence the phosphorylation status of Gad8. We have added the results in figure 1E. We cannot exclude the possibility that in higher concentrations or prolonged exposures to the drug, TORC2 will not be affected (as previously have been shown in the case of rapamycin). We have added the relevant material in the manuscript and in the figure legends to make this clear to the readers.

3. Figure 1E. Data do not fully support the authors' conclusions, i.e., changes in eIF2alpha and Maf1 phosphorylation upon treatment with RapaLink-1. The authors should replace these blots by other blots of better quality (lower exposure may help) or, at least, include semiquantitative measurements of band intensities.

To satisfy this point:

-We have repeated the experiments and western blots for phospho and total eIF2alpha and have replaced the panels.

-In addition, we have performed semiquantitative measurements of the ratios of the band intensities of phospho to total eIF2alpha and included the ‘normalised-to-corresponding-control’ numbers within the figure.

-For clarity, we have presented the maf1 results without the intense non-specific higher band and have labelled maf1 forms with lines and labels. The results show that mid and lower bands are more intense compared to the control state.

-We have also semi-quantified these results, at least for the middle band (one of the dephosphorylated forms of Maf1) and have inserted the numbers in the figure as in the case of the eIF2alpha forms.

-Following the suggestion of reviewer 4 (expert in TOR signalling in fission yeast), we have also tested for Psk1 phosphorylation as it might be easier to interpret, compared to Maf1. Together with rapamycin and rapalink-1 treatments at 2 hrs and 5 hrs timepoints we have also included results from

nitrogen starvation that strongly inhibit TORC1. The results are included in Figure 1F and show that following rapamycin and rapalink-1 treatments in these timeframes, Psk1-myc is significantly reduced. The effect of nitrogen starvation is extreme in these timepoints for Psk1-myc as well as for global protein content (we have also included Ponceau S as loading control).

-Following the suggestion of the reviewers we have also performed western blots for Gad8, a TORC2 target and have calculated the ratios of phosphorylated (P-Gad8) to non-phosphorylated (Gad8). We have included the 'normalised-to-corresponding-control' numbers within the figure (1E). The results show that Gad8 phospho- to non-phospho ratios do not change. We mention this in lines 108-111 and comment in lines 127-130. We, therefore, conclude that in these conditions, rapalink-1 affects primarily TORC1.

4. Figure 1I. In this manuscript, the authors extensively focus on comparing the effects of rapamycin and RapaLink-1. They should ideally assess the effect of both drugs in cells lacking FKH1 on the same experiment. Now, the authors perform the experiment with RapaLink-1 and rely on their own previous result (2013) for rapamycin.

We have performed the experiment with both rapalink-1 and rapamycin and directly compare the results obtained.

The panel is changed accordingly (now in Fig.1J), together with the corresponding results' section (lines 131-137).

5. To assess TORC1 activity in cells lacking agmatinases, the authors use a phospho-AKT antibody which, as they mention, is a readout for several AGC kinases. One of them is Gad8, which is a substrate of TORC2 (PMID: 24928510). Thus, the readout that the authors use to claim that lack of agmatinases activates TORC1 (Figure 1J) is not specific for TORC1. The authors should assess TORC1 in cells lacking agmatinases using more specific TORC1 readouts as described in Figures 1E and F.

To satisfy this point:

-We have amended the blot to include the signals corresponding to S6 kinases Sck1 and Sck2 (at 78 and 72 kD).

-We have performed repeated analyses (>6) for Maf1 in *wt* and in agmatinase mutant backgrounds. Nevertheless, the Maf1.pK does not perform well in the agmatinase mutants' background and the results have not been conclusive or easy to interpret (as also mentioned about this particular marker by one of our reviewers). For this reason, we have expanded our analyses and have now included readouts of phosphorylated eIF2alpha as in Figure 5G. Our results showcase that the phosphorylation levels of eIF2alpha are reduced in the agmatinase mutant backgrounds. We have provided semi-quantification of the corresponding intensities obtained through our western blot and have mentioned the results in lines 302-305.

The results agree with other data on cell growth/size of the *agm* mutants compared to *wt*.

6. Figure 5I. The authors should ideally compare the effect of single *agm1*, *tco89* and double *agm1 tco89* mutations on the same experiment (similar concern as in comment 4).

We have now repeated this experiment and have included the single *tco89Δ* data that the reviewer flagged as missing towards having this experiment complete with all appropriate controls. We have amended accordingly the figure legend and results section (lines 305-307).

7. Line 33. mTOR is a PI3 kinase-related kinase (also known as PIKK) and not a PI3-related kinase.

This is amended.

8. Related to Figure 3, lines 180 and 185. Figure 2B should be 3B. Figures 2C and 2D should be 3C and 3D.

This is amended.

REVIEWER 2

1. Avoid specific jargon, since this is a journal for non-specialized readers: Define rich YES media, introduce pik3, etc.

We have corrected points raised and wherever else we thought would be necessary (i.e. EMM2 media).

2. The authors show convincingly that Rapalink-1 lifespan extension depends on fkh1. But does it depend on autophagy? The experiment should be repeated using ATG knock outs. I acknowledge that the authors did not observe enhances autophagy upon Rapalink treatment but subtle changes in autophagic flux may have been overlooked by the assay used.

This is a fair enquiry from the reviewer regarding autophagy in this point and also within the next. Admittedly, the NGFP-Atg8 western blot does not showcase a huge change in the observed patterns between untreated and treated conditions. This is the reason we have stated at that part of the manuscript that autophagy changes are not pronounced (lines 98-99). Nevertheless, a gene ontology term enriched in upregulated genes following rapalink-1 treatment is catabolic process (Fig.S1D) and this includes genes related to autophagy. We mention more details on this in the next point. We have therefore amended our statements within the manuscript and mentioned more clearly the outcomes related to autophagy (lines 189-193).

84 genes are included in the ontology term: autophagy (GO:0006914) in fission yeast, within them 28 *atg* genes. We have thought about performing such experiments. Nevertheless, previous experience and the numerous genes involved in the process have led us to anticipate that single gene mutations will not affect the outcome of rapalink-1 lifespan extension due to compensations. In addition, performing CLS assays in the absence and presence of rapalink-1 for all the mutants (and in the appropriate biological replicates required), was not sensible, at least for a manuscript with a focus on agmatinases and their roles in TOR regulation.

Nevertheless, we did perform related CLS assays for some *atg* mutants including *atg2Δ*, *atg8Δ*, *atg12Δ*, *atg22Δ*. The results were as expected. Single *atg* deletions cannot repress the lifespan extension effect on rapalink-1, presumably, due to compensations. We have included these results in supplementary figure 2 and mention them in our results section (lines 192-197). A different way of assessing whether autophagy inhibition can restrict the lifespan extension effects of rapalink-1, was to combine rapalink-1 with autophagy inhibitors. Nevertheless, combinations with Chloroquine, Hydroxychloroquine or Bafilomycin A1 have been proven quite toxic.

3. Is spermidine enhanced upon Rapalink treatment? See Hofer et al, 2024, AUTOPHAGY? What about GABA concentrations?

The Hofer paper is a pivotal and extensive paper on the role of spermidine for fasting-mediated autophagy and longevity. Nevertheless, to assess for amounts of spermidine and GABA we would need to perform mass-spectrometry-based metabolomics experiments in appropriate repeats. Beyond the associated time (crucial in timely revising a manuscript) and the additional costs involved, these would provide considerable additional data to include and disseminate, that could deviate from the main theme of the current manuscript.

Interestingly, the *srm1* gene is not among the differentially expressed genes following rapalink-1 treatment: *srm1* codes for the unique spermidine synthase in *S. pombe*. Nevertheless, this does not mean that spermidine levels are not altered. Only that at the timeframe and rapalink-1 concentrations used, spermidine synthase is not induced at the transcriptional level.

Regarding autophagy: yes, most likely enhanced but not enough to be, strongly detected by the NGFP-Atg8 marker. Genes relating to catabolic functions are, obviously, upregulated following rapalink-1 treatment as shown in our gene ontology analyses.

In addition, using Angeli a gene ontology tool specifically developed for fission yeast, ~22% of the upregulated genes are related to catabolism (with 21 autophagy genes among them):

organonitrogen compound metabolic process	21.94%
---	--------

The gene signature of rapalink-1 upregulated genes is associate with genes upregulated when TORC1 is inhibited:

Caffeine and Rapamycin induced	34.8%
Nitrogen depletion total meiotic genes	22.57%

Among them, are genes related to autophagy.

In a transcriptional level, at least, it seems that autophagy genes are induced. We mention these points now in our manuscript (lines 192-193).

4. Also, the authors exclude this in the discussion: Does spermidine supplementation in Agmatine knock outs has the same effect as Putrescin supplementation on lifespan extension?

The reviewer asks to extend and discuss a specific point related to spermidine supplementations. However, we do not perform experiments with spermidine supplementation in this manuscript.

We believe the reviewer meant that we need to include in our discussion whether the supplementation of agmatine in *agm* mutants, has the same effect as putrescine supplementation, as these are the experiments we have performed in this paper.

We have now included related statements in our discussion. Supplementation of agmatine has more pronounced effects compared to putrescine and the reviewer was right to point this and request a reference within the discussion. The more pronounced effects of agmatine may relate to its multiple contribution to cellular health including mitochondrial fusion/fission balance (lines 362-364).

REVIEWER 3 (Charles Hoffman)

1. Supplementary Tables 1, 6 and 7 do not include common gene names, while Tables 2, 3, 4, and 5 do. Is there a reason for the difference? The addition of actual gene names would make these tables more accessible for the reader.

There was no reason for not including common gene names, this was an omission. We have now added the common names in the tables 1, 6 and 7.

We agree that this makes the information more accessible to the reader.

2. There are two sets of asterisks in Figure 1E that require explanation in the figure legend. I assume that for Maf1, these are presumed to represent phosphorylated and dephosphorylated forms, however the lower bands are so weak, I am not convinced that they are Maf1 versus a contaminating signal. What is the evidence that these bands represent dephosphorylated Maf1?

Are these bands missing from a Western of a *maf1* deletion strain? Does the protein shift to this lower band if the extract is treated with a phosphatase such as lambda protein phosphatase?

We apologise for this confusion. The upper strong band is a non-specific band representing a larger protein and we have now eliminated it from the presented image to avoid confusion.

We utilise a Maf1 V5-tagged strain previously used and published by the Petersen laboratory. It is not a Maf1-specific antibody that can be used in Maf1-deletion background in order to derive a relevant interpretation. The Maf1 signals are indeed weak, as previously reported, and discussed with the lab that has provided the strain. Nevertheless, the pattern observed is similar to published work and the tests requested by the reviewer have already been performed by the Petersen lab (the strain has been generated in) and included in their manuscripts.

3. The increase in CLS comparison in Figure 2H is discussed in absolute values. I suggest discussing this with a statistician to see whether it is more appropriate to assess the difference in the fold-increase. While this is still greater in wild-type cells (3.4 fold) than in *pik3D* cells (3.025-fold), I do not know if this is statistically significant or if that difference is important to any of the conclusions.

We have revisited the data and have repeated statistical testing as suggested by the reviewer. Indeed, while the difference is statistically significant, this is only marginal.

For this reason, we have decided to remove this piece of information for avoidance of confusion and for not making any strong statements regarding *pik3* in this context.

4. While not necessary, it would be nice for readers who are not experienced in SGA analyses to include a sentence in the Results section explaining what a negative interaction versus a positive interaction looks like in the assay.

We believe that the suggestion is important and valuable.

We have now included this information at the beginning of section to help the readers regarding positive versus negative genetic interactions (lines 262-271).

5. The authors need to clarify their use of asterisks with regard to statistical significance. Figure 1D has one, two and three asterisks without an explanation of their meaning. Elsewhere, they only use three asterisks and do explain that. Figure 5H seems like something that would warrant an assessment of statistical significance.

We have corrected the problem: Fig.1C and D have now only 1 asterisk, this is explained in the figure legend as $p < 0.01$. We have replaced all the asterisks in lifespans with the actual p values (or indications that the p values are < 0.01) and have mentioned within the figure legend that these correspond to log rank tests.

The original 5H graph has been removed as we have now included additional western blots following reviewers' suggestions.

6. For Figure 5H, are these technical or biological replicates?

These are biological replicates.

7. Minor typos

Line 177- remove open parenthesis

Corrected.

Line 221- oxygen-containing compounds

Corrected.

Line 253- replace “to show” with “showing” to create parallel structure with “exhibiting”

Replaced, currently line 280.

Line 277- “seems is enhanced”- please clarify this sentence

The data have been amended following reviewers' comments with western blots added. The sentence has been changed for clarity (currently, line 302).

Lines 280-281- “but does not reflect, in this case, to the ageing phenotype”- please edit

The phrase is removed to avoid confusion.

Line 294- ensures

Corrected, currently in line 317.

REVIEWER 4

1. In Fig. 1E, TORC1 activity was monitored after 2 and 5 hours of rapamycin as well as rapalink-1 treatment. However, as the mitotic index is significantly increased even after 30 min of treatment with TORC1 inhibitors (Fig. 1A and B), the authors should assess the TORC1 activity at earlier time point (e. g. 30 min after treatment). Previous literatures have demonstrated that Maf1 is effectively dephosphorylated in the presence of rapamycin (Du et al., 2012, Biol. Open, 1, 884; Morozumi et al., 2021, J. Cell Sci. 134, jcs258865). On the other hand, only a tiny portion of Maf1 was dephosphorylated in authors' experiments, suggesting a possibility of mild suppression of the TORC1 activity. To carefully assess the TORC1 activity in these experimental conditions, another TORC1 substrate, such as Psk1 (much easier to detect its phosphorylation state than Maf1 (Nakashima et al., 2012, J. Cell Sci. 125, 5840; Morozumi et al., 2021, cited above)) should be examined.

This is a very good suggestion. We have performed the additional western blots using the Psk1-myc strain that have been reported previously. Our results demonstrate that Psk1-myc is drastically reduced especially in the five hours timepoints following rapamycin and rapalink-1 treatments (data included in Figure 1F). The difference in our results and the already published (regarding phospho- and dephosphorylated ratios of Psk1-myc) might be the timeframes used. In this manuscript, we wanted to use the same timeframes (2 and 5 hrs) as used with all the other western blot markers analysed.

Unfortunately, at a 30min timepoint we do not see effects on the various markers. It might therefore be that different TOR-dependent functions have different kinetics/characteristics and do not change in the same timeframes. E.g. while the mitotic index changes at 30 min, eIF2alpha remains unchanged (even after 1 hrs), but changes at the 5 hrs timepoint.

2. The authors state that the up regulation of genes related to vacuolar transport and organization by rapalink-1 resembles the observation treated by torin1, an ATP-competitive inhibitor of both TORC1 and TORC2 (p8, lines175-177). Thus, it is likely that rapalink-1 is able to inhibits not only TORC1 but also TORC2 in fission yeast. Indeed, rapalink-1 can suppress both mTORC1 and mTORC2 activity in mammalian cells (e.g. Rodrik-Outmezguine et al.,

2016, Nature 534, 272). Thus, authors should examine whether rapalink-1 inhibits TORC2 in fission yeast by monitoring the phosphorylation of TORC2 substrate such as Gad8.

We appreciate the input from all reviewers and this is a point that has been raised elsewhere too.

We have examined Gad8 phosphorylation levels vs the unphosphorylated one using a published strain we have used before, PMID: 23551936.

Our results show that rapalink-1 treatment, at the concentrations and the timepoints examined, does not influence Gad8 phosphorylation (Figure 1E).

See also reviewer 1, point 2.

3. The RNA-seq analysis shown in Fig. 3 revealed that the expression of agmatinase genes are increased upon rapalink-1 treatment. As the authors focused on these genes and performed further experiments, this observation should be confirmed by individual RT-qPCR.

This is a useful validation requested by the reviewer.

We have performed the required qPCRs for all 3 agmatinase genes and included these in Figure 3G.

Our results validate the RNAseq data. We have included all relevant information in materials and methods (lines 440-450) and our results' section (lines 210-213).

4. In Fig. 5G, although the authors visualized the phosphorylation of TORC1 target kinases by using anti p-Akt targets antibody, it remains possible that those obtained bands are just non-specific signals. The authors should eliminate this possibility by including the control experiment under TORC1 inhibition, such as nitrogen starved or rapamycin-treated conditions.

We have now amended the western blots, and we have performed additional western blots as suggested by the other reviewers.

The results demonstrate that agmatinase mutants have reduced phospho-eIF2alpha levels pointing to the suggestion of increased translation.

5. By combining the results shown in Fig. 3G-H and Fig. 5, the authors proposed that shorter lifespan of the agm mutant cells compared to wild-type is due to the increased TORC1 activity. To further confirm this possibility, lifespan of the amg mutants in the presence of TORC1 inhibitor such as rapamycin and rapalink-1 needs to be tested.

We have performed the CLS assays on *agm* mutants following rapalink-1 treatment as requested by the reviewer.

We have provided the asked additional lifespan in the new Supplemental figure 4. Our results show that the lifespan of the individual *agm* mutants is prolonged following drug treatment, probably due to compensation between the agmatinases. The lifespan extension for *agm2* is less pronounced compared to the ones observed for *agm1* and *agm3*. Nevertheless, the change is significantly different in all cases (log rank p value <0.01).

We were unable to obtain a triple *agm* mutant, or a double in which *agm2* is deleted (we do show lifespan data for *agm1Δ agm3Δ* in Figure 3K).

This was possibly due to lethality.

Minor points

6. Fig. 1A, B: Dot lines are too thin, being difficult to see (particularly in rapamycin treated cells). Thicker lines should be used.

We have now included black thicker lines within the figure and in both A and B panels to help towards this.

7. In Fig. 1E, the authors demonstrated that rapamycin and rapalink-1 treatment did not induce autophagy by monitoring the cleavage of GFP from Atg8. This result is consistent with previous observations (Takahara et al., 2012 *Genes Cells*, 17 698; Morozumi et al, 2021 cited above). However, the authors state autophagy may be up regulated (line 98-). This does not reflect the observation in Fig. 1E, and thus, should be eliminated.

We have amended the statement. Other reviewers' requested more details about autophagy (see reviewer 2). The statement above is amended as following: 'Examination of GFP-Atg8 patterns following drug treatment in the aforementioned time points, do not show a significant degree of processing indicating that effects are not pronounced, at least in these timeframes'. We then present and discuss the transcriptional effects on autophagy genes and perform lifespan assays for *atg* mutants as requested. Please also see reviewer 2 points 2 and 3.

8. Fig. 1E: Information about what asterisks indicate should be included.

We have amended the Figure according to suggestions from all the reviewers.

9. Line 121: The P value of the result in Fig. 1H is "0.047" in the main text and figure legend, while "0.054" in the figure. Which is true?

We apologise for the confusion. We have now corrected this. 0.054 was the right figure.

10. Lines 174, 180, 185: Cited figures should be Fig. 3 but not Fig 2.

This is corrected.

11. Line 177: Parenthesis should be eliminated.

This is corrected.

12. Fig. 5A: Please mention what the blue and red lines indicate.

This is mentioned within the main text and the figure legend.

13. Among the kinases listed in line 277, Gad8 is a direct substrate of TORC2 but not TORC1, and thus, should be removed from the sentence.

This is corrected.